# Neural Networks for Financial Time Series Forecasting

**DOI:** 10.3390/e24050657

**Published:** 2022-05-07

**Authors:** Kady Sako, Berthine Nyunga Mpinda, Paulo Canas Rodrigues

**Affiliations:** 1African Institute for Mathematical Sciences (AIMS)-Cameroon, Limbe P.O. Box 608, Cameroon; kady.sako@aims-cameroon.org (K.S.); bmpinda@aimsammi.org (B.N.M.); 2Department of Statistics, Federal Universityof Bahia Salvador, Salvador 40170-110, Brazil

**Keywords:** Recurrent Neural Networks (RNNs), Artificial Neural Networks (ANNs), stock market indices data, Long Short-Term Memory (LSTM), Gated Recurrent Unit (GRU), currency exchange rates

## Abstract

Financial and economic time series forecasting has never been an easy task due to its sensibility to political, economic and social factors. For this reason, people who invest in financial markets and currency exchange are usually looking for robust models that can ensure them to maximize their profile and minimize their losses as much as possible. Fortunately, recently, various studies have speculated that a special type of Artificial Neural Networks (ANNs) called Recurrent Neural Networks (RNNs) could improve the predictive accuracy of the behavior of the financial data over time. This paper aims to forecast: (i) the closing price of eight stock market indexes; and (ii) the closing price of six currency exchange rates related to the USD, using the RNNs model and its variants: the Long Short-Term Memory (LSTM) and the Gated Recurrent Unit (GRU). The results show that the GRU gives the overall best results, especially for the univariate out-of-sample forecasting for the currency exchange rates and multivariate out-of-sample forecasting for the stock market indexes.

## 1. Introduction

Finance is one of the most dynamic and valuable sectors that plays a vital role in the economic development of a country [1]. The behavior of currency exchange rates or stock market indexes is a good indicator of the financial health of a country, and is used by investors to maximize their gains in the financial market. A financial market represents a marketplace where currencies, bonds, equity, and securities are traded. Generally, in a financial market, the stock market allows investors to sell and buy public companies’ shares while the forex market allows investors to trade currencies. Throughout the world, the forex market constitutes the most liquid financial market. Aiming at profiting, every investor closely follows the evolution of the financial market in order to elaborate good strategies for better investment decision making. Therefore, it becomes crucial for any investor to be aware of current political decisions and to understand how past behaviors can impact the future financial market movement.

Much research has been devoted to time series forecasting with various techniques proposed, from the more classical statistical or traditional models such as the autoregressive integrated moving average (ARIMA) model and its variants, autoregressive conditional heteroskedasticity (ARCH) and generalized autoregressive conditional heteroskedasticity (GARCH), or the exponential smoothing and its adaptations [2], to more complex models based on deep learning. Although the ARIMA and other linear models are usually more effective in short-term forecasting for univariate data [3,4,5], these linear models trend to fail to predict with higher accuracy the behavior of the stock market over time because of its complexity. One of the alternatives to overcome this limitation has been the deployment of one special type of Artificial Neural Networks (ANNs) called Recurrent Neural Networks (RNNs). The particularity of these ANNs models is their ability to capture important patterns from complex and non-linear time series, to obtain more accurate predictions [6,7,8,9]. Other applications of ANNs for time series forecasting can be found in [10,11,12,13].

Financial time series forecasting is one of the most challenging tasks in time series forecasting due to the influence of social, political, and economical factors that help define the stock market behavior. Due to their importance in finances and also for investors, stock market forecasts and currency exchange rates forecasts have received much attention in the time series literature. In [14], a systematic review for deep learning studies and applications for stock market forecasting is presented, and four main points of view are discussed: predictor techniques, trading strategies, profitability metrics, and risk management. Their methodology selected 34 papers, concluding that the LSTM (including hybrid methodologies using the LSTM) is the most applied neural network representing 73.5% of the analyzed publications. The article [15] presented a survey of stock market forecasting using computational intelligence approaches such as artificial neural network, fuzzy logic, genetic algorithms and other evolutionary techniques. In terms of techniques for decision aid, ref. [16] proposed a method to identify how multicriteria decision aid can assist the investment portfolios formation, increasing the reliability of decision making, ref. [17] proposed a multicriteria optimization approach for the stock market feature selection, and ref. [18] proposed a method based on information reliability and criterion non-compensation to manage stock selection problems. When dealing with currency exchange rates, also several studies and strategies have been proposed. In particular, ref. [19] provided a review of applications of ANN to forecast foreign exchange rates, and ref. [20] presented a comprehensive review of 82 soft computing hybrid methodologies, including ANN, evolutionary computation, fuzzy logic, support vector machine and chaos theory for currency exchange rates. For example, ref. [21] proposeda forecasting method based on an RNN and a convolutional neural network. A more comprehensive systematic literature review between 2005 and 2019 for time series forecasting with deep learning was presented by [6].

The main objective of this work is to assess the performance of the Recurrent Neural Network (RNN) and its two variants: Long Short-Term Memory (LSTM) and Gated Recurrent Unit (GRU) in the context of closing price forecasting for both stock market indexes and currency exchange rates. Besides the comparison between versions of recurrent neural networks for financial time series forecasting, this work includes time series from stock market indexes and currency exchange rates from some of the most important economies from four continents, which allows us to make a wider analysis and comparison between world regions. The forecasting models are evaluated using metrics such as the Root Mean Square Error (RMSE) and the Mean Absolute Error (MAE). The models were trained and evaluated on daily stock data of eight stock market indices and six currency exchange rates from countries in four continents. The data was collected from 2 January 2008 to 28 May 2021. The selected stock market indices considered in this study are: New York Stock Exchange (NYSE), National Association of Securities Dealers Automated Quotations (NASDAQ), Johannesburg Stock Exchange (JSE), Nigeria Stock Exchange (NSE), Euronext, Frankfurt Stock Exchange (FRA), Shanghai Stock Exchange (SSE) and Japan Stock Exchange (JPX), and the selected currency exchange rates considered are: South African Rand (ZAR/USD), Nigeria Naira (NGN/USD), British Pound (GBP/USD), Euro (EUR/USD), Japanese Yen (JPY/USD), and Chinese Renminbi (RBM/USD). Univariate and multivariate models are considered.

This paper is structured as follows: The data and recurrent neural networks methodology for time series forecasting are presented in Section 2. The results and discussions are presented in Section 3, and the concluding remarks end the paper in Section 4.

## 2. Materials and Methods

This section presents a contextualization and description of the two data sets: (i) stock market indexes; and (ii) currency exchange rates, together with a detailed description of the forecasting models used in this paper.

### 2.1. Data

The data considered in this paper includes two databases: (i) the daily stock data of eight stock market indices: the NYSE Composite (NYA) and NASDAQ Composite (IXIC) from America, the Euronext 100 (N100) and Deutsche Boese DAX index (GDAXI) from Europe, the SSE Composite index (000001.SS) and Nikkei 225 (N225) from Asia, and the Global X MSCI Nigeria ETF (NGE) and FTSE/JSE Africa index (J580.JO) from Africa; and (ii) the daily exchange rate of six currencies, in reference to the United States dollar (USD): South African rand (ZAR/USD), Nigeria Naira (NGN/USD), British Pound (GBP/USD), Euro (EUR/USD), Japanese yen (JPY/USD), and Chinese Renminbi (RBM/USD). The data was collected between 2 January 2008 and 28 May 2021, except for the two African currencies that were only available from 3 April 2013. There are some differences in the number of observations for each time series (see Table 1 and Table 2), with, e.g., J580.JO index having 51 missing values. Each data has six features which are: the High, Low, Close, Volume, Adjust Close and Open values. In this paper, we will focus on forecasting only the: (i) closing stock price for the eight stock indices, and (ii) closing prices of the six currency exchange rates. Both data sets were obtained from Yahoo Finance.

### 2.2. Time Series Analysis

A time series is a sequence of observations recorded at a regular time interval. Depending on its frequency and objective, the observations can be taken hourly, daily, monthly, annually, or any other. For instance, daily stock market prices, daily currency exchange rates, hourly temperatures in degrees Celsius, quarterly sales results of a company, etc.

An observed time series can be decomposed into four main components and each of them expresses a particular aspect of the movement of the time series observations. These components are: (i) trend, an increasing or decreasing behavior of the series over time; (ii) seasonality, the repeating patterns or cycle behavior of the series over time; (iii) cyclic, repeating patterns of the series over time but, unlike seasonality, it is not of a fixed frequency; and (iv) noise, also called error component or the residual, is a random irregularity found in the time series that can not be explained.

Decomposing a time series into components is a type of preliminary time series analysis that provides meaningful information about the characteristics of the time series in order to predict its future behavior based on the understanding of its past observations, and can be applied to all fields of application where data is collected along time.

In this paper, we are interested in financial market analysis and forecasting, in particular stock market indexes and currency exchange rates. This means that we analyze the past movement of the time series to make a future prediction. There are several machine learning (ML) models that have been proposed for stock market and exchange rate forecasting. However, since we are working on sequential data, we will be focused on a special type of artificial neural networks (ANNs) called the recurrent neural network (RNN), and its variants: the long short-term memory (LSTM) and the gated recurrent unit (GRU).

Another important component to be considered in time series forecasting is the case of when exogenous explanatory variables are available and can be used as covariables to help improve forecasting. In this paper, we consider univariate models that deal with the response variable of interest (the close stock market indexes and currency exchange rates), and multivariate models that also consider the exogenous explanatory variables: high, low, open, and closed.

In the next subsections, we introduce some basic concepts about the recurrent neural networks used in this paper, and provide key references to relevant literature.

### 2.3. Artificial Neural Networks

Artificial neural networks (ANNs), also called neural networks (NN), constitute a subset of machine learning, more precisely deep learning (DL), which is based on numerical algorithms that simulate the behavior of biological systems composed by neurons. It is made up of numerous interconnected unit cells called artificial neurons (ANs), that are composed of three components: an input, an activation function, and an output, that can be defined as follows:Input: It represents a matrix array *X* of several data types such as image, audio signal, data frame etc. These input values are then processed as a linear combination of the latter with a weight matrix *W* and a bias vector. The weight is the additional parameter of a neural network that transforms the input data in the hidden layers of the network. It decides the speed at which the activation function will fire. The bias is a parameter of the neural network that helps the model to best fit the given data.Activation function: The process input is fed into an activation function defined as σ(WX+b) which triggers a signal whenever a condition is satisfied. The four common types of activation functions are: the Rectified Linear Unit (ReLU), Sigmoid, Tangent hyperbolic (tanh), and Softmax. They are defined, respectively, in Equations (Equation 1)–(Equation 4).
(1)$$\begin{array}{ccc} ReLU(k) & = & \max(0,k) \end{array}$$
(2)$$\begin{array}{ccc} Sigmoid(k) & = & \frac{1}{1+e^{-k}} \end{array}$$
(3)$$\begin{array}{ccc} \tanh(k) & = & \frac{e^{k}-e^{-k}}{e^{k}+e^{-k}} \end{array}$$
(4)$$\begin{array}{ccc} Softmax_{i}\left(k\right) & = & \frac{e^{k_{i}}}{\sum_{j}e^{k_{j}}},\;\;\forall i=1,2,\cdots ,n. \end{array}$$Output: It is the numerical result after fitting the input into the activation function.

The architecture of a ANN with one hidden layer is presented in Figure 1.

Deep Neural Networks (DNNs) are a type of ANNs constructed from multiple layers connecting the input to the output. The three building blocs of DNN are: the input layer, the hidden layer, and the output layer which are presented in Figure 2.

In deep learning, the ultimate goal is to find the set of parameters (W,b) that best predict the output. This is performed on a three-step learning process.

The first step is based on the forward propagation approach. That is, the input is fitted into the input layer of the DNN and an output is obtained as defined in Equation (Equation 5)
(5)$$\begin{array}{c} {\hat{Y}}^{\left(l\right)}=\sigma \left(W^{\left(l\right)}X+b^{\left(l\right)}\right), \end{array}$$
where ${\hat{Y}}^{\left(l\right)},W^{\left(l\right)}$ and $b^{\left(l\right)}$ are, respectively, the output, the weight and the bias of layer *l*.

The second step consists of comparing the computed output with the true output by means of a cost function (loss function). “Cross-entropy” or “log loss” and the mean squared error (MSE) constitute the two main types of loss functions when we are training neural network models. These are defined in Equation (Equation 6) (for the classification problem) and in Equation (Equation 7) (for the regression problem).
(6)$$\begin{array}{ccc} Cross-entropy & = & -\frac{1}{N}\sum_{i=1}^{N}\sum_{j=1}^{m}Y_{ij}\cdot \log\left({\hat{Y}}_{ij}\right) \end{array}$$
(7)$$\begin{array}{ccc} MSE & = & \frac{1}{2N}\sum_{i=1}^{N}{\left({\hat{Y}}_{i}-Y_{i}\right)}^{2}, \end{array}$$
where *N* and *m* are, respectively, the number of input entries and the number of different classes. $Y_{ij}$ and ${\hat{Y}}_{ij}$ are, respectively, the true value and the predicted value for the *i*th input entry and *j*th class.

The last step consists of using “optimizers” to update the weights and the bias. Optimizers are algorithms used to modify some parameters of the neural network such as weights and learning rate, in order to minimize loss functions and provide accurate results. The learning rate is a tuning parameter or hyper parameter that determines the step size at each iteration while moving to a minimum of a loss function. The most popular optimizer used by a neural network is the gradient descent. The set of these three steps is called the backward propagation or back propagation.

Now, after these introductory concepts, we are able to introduce and give details about the Recurrent Neural Network (RNN) and its variants: the Long Short-Term Memory (LSTM) and the Gated Recurrent Unit (GRU).

### 2.4. Recurrent Neural Networks (RNNs)

The Recurrent Neural Network (RNN) is one of the promising ANNs that has shown accurate results for time series forecasting. It is made up of a series of interconnected neural networks at different time intervals or time steps. The simple RNNs have the ability to remember most of the past information in order to predict the current and the future information, being the output of the previous step fed as the input of the current step. An example of an RNN can be seen in Figure 3.

Despite its usefulness for time dependent data, the simple RNNs are exposed to the problem of vanishing and exploding gradient which arises because of the repeated multiplication of gradients that can either increase or decrease based on the number of layers of the neural network. To counteract this problem, ref. [22] proposed a model capable of remembering and forgetting some of the past information in order to predict the current and the future information: the Long Short Term Memory (LSTM). LSTMs are built from compound series of repeated cells, and are made up of three building blocks: the “input gate” (also called the update gate, makes decisions based on relevant information to update the current cell state), the “forget gate” (to pay attention to which signal to be deleted from the memory cell for the current step) and the “output gate” (determines the next value of the hidden state). Complete details about the LSTM can be found in [22].

The LSTM is one of the powerful RNNs that can be used for financial time series forecasting [6,9]. However, it has many parameters to be set during the training phase, which requires a large computational power, making the data processing slow. To help solve this problem, a recurrent neural network called the gated recurrent unit (GRU) was proposed by [23]. The GRU architecture consists of a “reset gate” (responsible for the amount of information to be forgotten during data processing) and an “update gate” (responsible for the amount of transmission of the input and the previous output to the next cell). The GRU also has a very important cell called the current memory, whose role is to ensure that all important information is transmitted to the next cell. More details about the GRU can be found in [23].

### 2.5. Accuracy Measures

Two accuracy measures were used to evaluate the train, validation and forecasting error: (i) the Root Mean Squared Error (RMSE); and (ii) the Mean Absolute Error (MAE). To evaluate the out-of-sample forecasts, we considered the last *l* observations and the RMSE, and MAE can be written as: (8)$$\begin{array}{c} RMSE=\sqrt{\frac{1}{l}\sum_{i=N-l+1}^{N}{\left(Y_{i}-{\hat{Y}}_{i}\right)}^{2}}, \end{array}$$(9)$$\begin{array}{c} MAE=\frac{1}{l}\sum_{i=N-l+1}^{N}\left|Y_{i}-{\hat{Y}}_{i}\right|, \end{array}$$
where $Y_{i}$ are the last *l* observed values and ${\hat{Y}}_{i}$ are the predicted values.

Having discussed the fundamental concept of RNN models, we move forward to present the results obtained using these univariate and multivariate RNN models to forecast the closing price of the eight stock indices and six currency exchange rates. Moreover, based on the RMSE and MAE, we present a comparative analysis to evaluate the performance of the different models to forecast the eight stock market indexes and the six currency exchange rates.

## 3. Results and Discussion

In this section, the historical data of eight stock indices and six currency exchange rates will be analyzed. These data are used to compare the performance of the univariate and multivariate versions of the three neural networks under consideration: (i) the simple RNNs; (ii) the LSTM; and (iii) the GRU.

### 3.1. Preliminary Analysis

Table 1 and Table 2 present the summary statistics of the closing price for the eight stock market indices and for the six currency exchange rates, respectively, including number of observations, maximum, minimum, mean and standard derivation.

Figure 4 and Figure 5 show the behavior of the closing prices of the eight stock market indices and six currency exchange rates along the time (days), where significant differences in behavior are visible.

### 3.2. Model Fit

#### 3.2.1. Data Splitting

To design a prediction model for the eight stock indices, and the six currency exchange rates, using the RNN algorithms, we start by splitting the data into train, validate and test sets, for each index and currency exchange rate.

The train set was considered to represent 80% of the whole data, and the test data contains 20%. The validate set constitutes 20% of the train set. Figure 6 and Figure 7 illustrate the data splitting for a time series in each data set. The validation set will be used to evaluate the performance of the three models under consideration, during the training process.

#### 3.2.2. Features Selection

To predict the future values of the closing price using RNNs models, univariate and multivariate forecasting algorithms can be used. The univariate time series forecasting consists of the use of historical data of the closing price to predict the future values of the closing price. However, as it is well known, the stock market indexes and the currency exchange rates are volatile and complex with movements and behaviors that are constantly changing, which makes the univariate forecasting models likely to be very limited and prone to error as they reduce the complexity of the index to a single variable [24]. Therefore, in this analysis, besides the use of the univariate forecasting models, we also consider the multivariate forecasting models, which take into account exogenous explanatory variables such as the daily observations for high, low, open and closing price, to forecast the future behavior of the closing price.

The feature selection was performed as presented in Figure 8. Two variables were not considered: (i) the adjust closing price (Adj Close) because it has the same value of the closing price; and (ii) the “volume” because it is unknown for the currency exchange rates and we aimed at comparable comparisons for both data sets.

#### 3.2.3. Data Pre-Processing

Data pre-processing is one of the most important and often difficult tasks to perform whose main objective is to transform the raw data into a comprehensible format, that is, to encode the raw data into a form that can be easily analyzed and interpreted by the algorithm. Non pre-processed data usually leads to a poor model with low accuracy.

From Figure 4 and Figure 5, large ranges of features can be observed. In that way, there is a need to normalize the data before feeding the data to the models. The Python MinMaxScaler function was used to scale the range of the data between zero and one, with its equation being written as
(10)$$\begin{array}{c} X_{norm}=\frac{\left(X-X_{min}\right)}{\left(X_{max}-X_{min}\right)} \end{array}$$
where $X_{norm}$ is the output of a feature *X* (e.g., high, close, open, low) after performing the normalization, $X_{min}$ and $X_{max}$ are the minimum and the maximum of the feature *X*. To use deep learning techniques, we need to do some other transformations on the time series data. Indeed, the RNN and its variants use supervised techniques to fit a model. So, we must re-frame the time series data into a supervised learning data-set, with inputs and outputs from the historical data.

The sliding window approach was performed to transform the time series data (train, validate, and test sets) into a three-dimensional data. The first dimension is called the sequence or the tensor, the second is the number of time steps or the number of lags, and the third is the observation at a specific time step called feature. The set of this three-dimensional is called mini-batch. For instance, let us consider the historical data of the N100 index with 3425 observations and five features. Using the sliding window technique with five features and five as a number of time steps, the output of the sliding will give a three-dimensional array which contains 3420 sequences and each sequence will contain five time steps and five features. So, the 3420 sequences constitute the input entries and the five features constitute the outputs for the supervised learning problem.

The missing values in the J580.JO stock data were replaced by the median of each column of the data. We used the median to deal with this problem because of its robustness in dealing with potential outliers.

#### 3.2.4. Parameters Tuning

Parameters tuning or the finding of the best parameters to accurately forecast the closing price is one of the most important tasks in predictive modeling.

It can be observed that, as the size of the time series increases, increasing the number of neurons can improve the result especially for the LSTM and GRU models. The right parameterization of the learning rate is also necessary to make good predictions. The use of regularizers helps to avoid or decrease the degree of over-fitting. We should also keep in mind that, after obtaining the fit/forecast values, they should be scaled back to the original measurement unit, i.e. we must reverse the time series values after forecasting with the RNN models for a more objective and fair comparison.

#### 3.2.5. The Complete Algorithm and the Results for Model Fit

In this subsection, we summarize the procedure described in the section, from the data collection to the accuracy measurement of the forecast models. The diagram for the algorithm/procedure can be found in Figure 9.

Table 3 presents the full specification of the parameters for the simple RNN, LSTM and GRU models.

Table 4 and Table 5 show the RMSE for the model fit (train and validation sets) based on the models with parameters defined in Table 3. The mean absolute values are in Table A1 and Table A2 of the Appendix A.

Table 6 and Table 7 show the RMSE for the model fit (train and validation sets) based on the models with parameters defined in Table 3. The mean absolute values are in Table A3 and Table A4 of the Appendix A.

### 3.3. Model Forecasting

In this section, we present and discuss the results for model forecasting of the univariate and multivariate RNN, LSTM and GRU models. The RNN model is made up of an input layer followed by either a simple RNN, LSTM or GRU layer, and will end with a dense output layer. For each index, the number of inputs, layers, and activation functions (hyperbolic tangent) are the same for all models in order to make consistent comparisons.

The RNN, LSTM and GRU models have been executed for five look-back, which constitute an element of the input set with the number of features. After fitting the model, we obtained forecasts for the test data, for both data sets and for both univariate and multivariate time series forecasting, as defined before, being the performance of the three models evaluated using the RMSE and the MAE.

Table 4 and Table 5 show the RMSE for the model forecasting (test set) based on the models with parameters defined in Table 3. The mean absolute values are in Table A1 and Table A2 of the Appendix A.

Table 6 and Table 7 show the RMSE for the model forecasting (test set) based on the models with parameters defined in Table 3. The mean absolute values are in Table A3 and Table A4 of the Appendix A.

As it can be observed in Table 1 and Table 2 and in Figure 4 and Figure 5, the stock market indexes and currency exchange rates show very different behaviors along time, which would possibly require different inputs and parameter specification. In this case, for comparative purposes, we consider the same model specification for all time series. By making an overall comparison between the results for all time series based on the RMSE and MAE, most of the predicting models (univariate and multivariate) satisfy the following equations:$$\begin{array}{c} RMSE_{test}\leq RMSE_{validate}\leq RMSE_{train}\\ and\\ MAE_{test}\leq MAE_{validate}\leq MAE_{train} \end{array}$$
This means that, the RNN models are fitting relatively well (not overfitting not undeviating) the stock market indexes and currency exchange rates.

There is a strong correlation between the time series that are used for the multivariate models (close, low, open and high), which results in similar values of RMSE and MAE for the univariate and multivariate forecasts.

Based on the RMSE and MAE of the test set, it can be concluded that the GRU model is the overall best model, especially for the univariate out-of-sample forecasting for the currency exchange rates and multivariate out-of-sample forecasting for the stock market indexes.

It should be kept in mind that only five days in the past were used to build the forecasting models (for the three RNN models), which is considered as a short time step and explains why the LSTM was less efficient. It is known that the LSTM model is more efficient in long time step forecasting. The plots presented in Figure A1 and Figure A2 of the appendix provide a graphical comparison between the performance of the three recurrent neural networks, considering the univariate and multivariate algorithms, for one currency exchange rate and one stock market index. From the analysis of these results, it appears that the RNN models manage to capture important patterns of stock prices allowing them to make an accurate prediction.

All analyses were performed using Python. The scripts used in this manuscript are available upon request from the corresponding author of this paper.

## 4. Conclusions and Further Work

The aim of this paper was to assess the performance of the neural network models, particularly the recurrent neural network models, for financial time series forecasting. To achieve our objectives, we compared the performance of three recurrent neural network models, the simple recurrent neural network, the long short-term memory, and the gated recurrent unit. We performed a feature selection before conducting data pre-processing, followed by parameters tuning on the train set and computation of the out-of-sample forecasts on the test set. The univariate and multivariate algorithms were used in two data sets: eight stock market indexes, and six currency exchange rates.

Based on the analysis and comparisons using the RMSE and MAE, and based on the model specification, it could be concluded that the GRU model is the overall best model, especially for the univariate out-of-sample forecasting for the currency exchange rates and multivariate out-of-sample forecasting for the stock market indexes.

For future directions, it would be interesting to use recurrent neural networks to correlate different stock market prices and currency exchange rates, taking into account their economic, political and socio-cultural differences, and also to consider pre-processing methods such as the singular spectrum analysis [25,26,27] before feeding the recurrent neural networks. The combination of deep learning models with technical and fundamental analysis of financial markets can also be considered to help investors make better decisions.

## Figures and Tables

**Figure 1 entropy-24-00657-f001:**
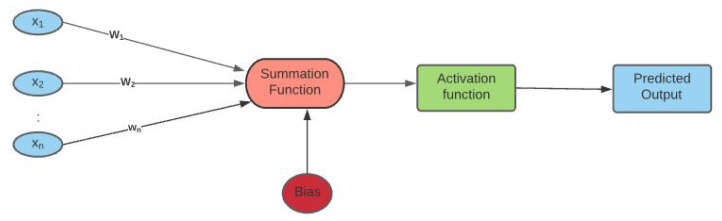
The input data $\left(x_{1},x_{2},\ldots x_{n}\right)$ are processed as a linear combination with weights $\left(W_{1},W_{2},\ldots W_{n}\right)$ and bias *b*. Then, the inputs are fed into an activation function in order to get the final result (predicted output).

**Figure 2 entropy-24-00657-f002:**
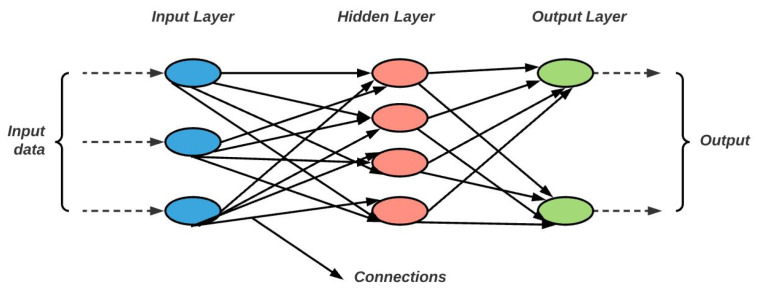
The input layer transmits the initial data to the hidden layers through the connections (the black arrows) between them. After processing, the hidden layers transmit the result with the help of connections to the output layer.

**Figure 3 entropy-24-00657-f003:**
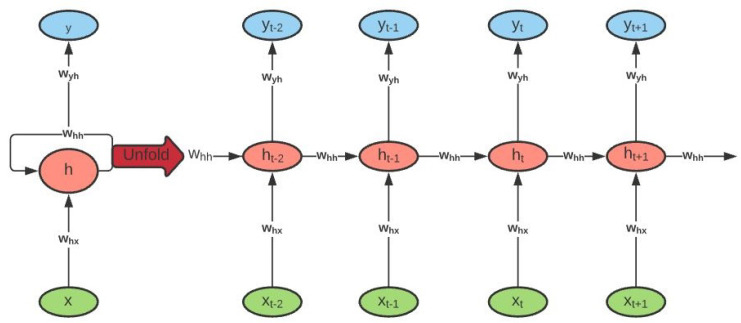
Unlike other neural networks, the RNN uses the same parameters for the input entries because it does the same task on all inputs to produce the output. This reduces the complexity of the parameters.

**Figure 4 entropy-24-00657-f004:**
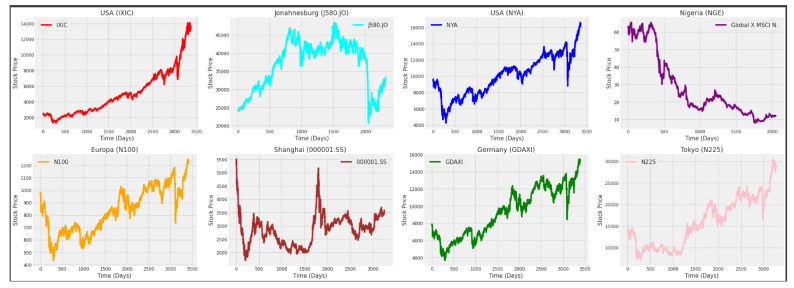
Closing price of the stock market indices per day. From left to right: American indices (IXIC and NYA) and African indices (J580.JO and NGE) [top], and European indices (N100 and GDAXI), and Asian indices (000001.SS and N225) [bottom].

**Figure 5 entropy-24-00657-f005:**
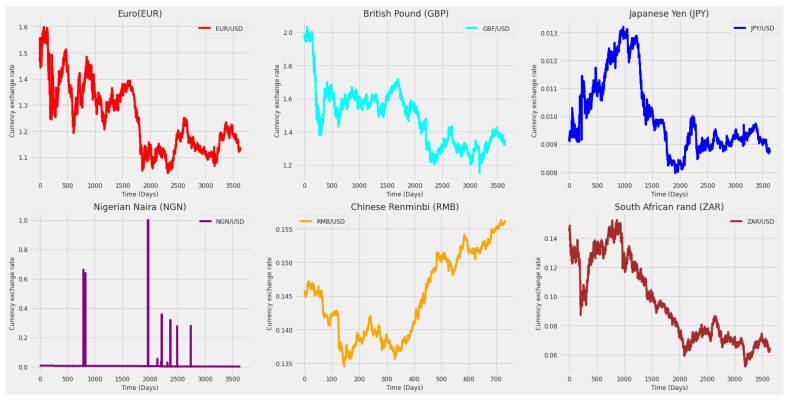
Closing price of the currency exchange rate (related to the USD) per day. From left to right: Euro, British Pound and Japanese Yen [top], and Nigerian Naira, Chinese Renmindi and South Africa Rand [bottom].

**Figure 6 entropy-24-00657-f006:**
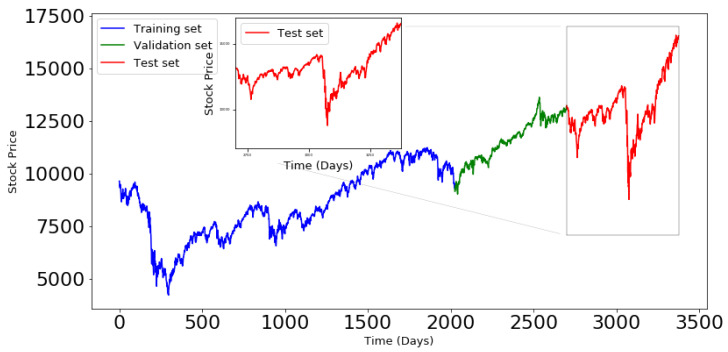
Train, Validate and test sets for the NYA index.

**Figure 7 entropy-24-00657-f007:**
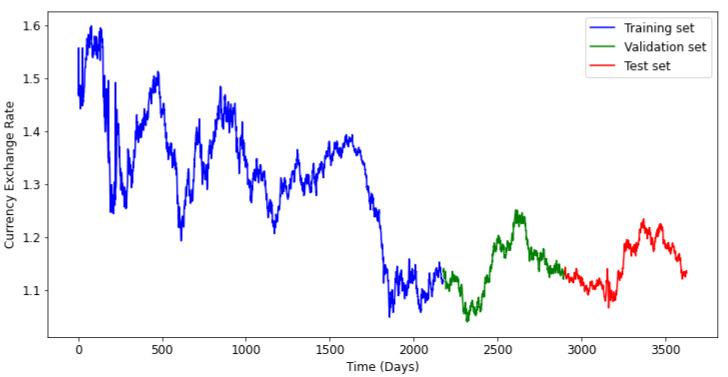
Train, Validate and test sets for the Euro (EUR/USD).

**Figure 8 entropy-24-00657-f008:**
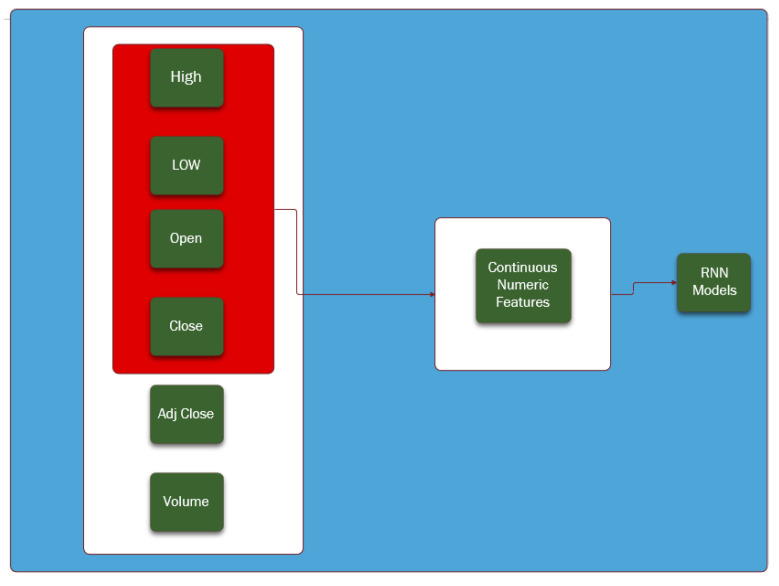
Feature selection considering the numeric variables: High, Low, Open, Closed and Volume, used to train the RNN models.

**Figure 9 entropy-24-00657-f009:**
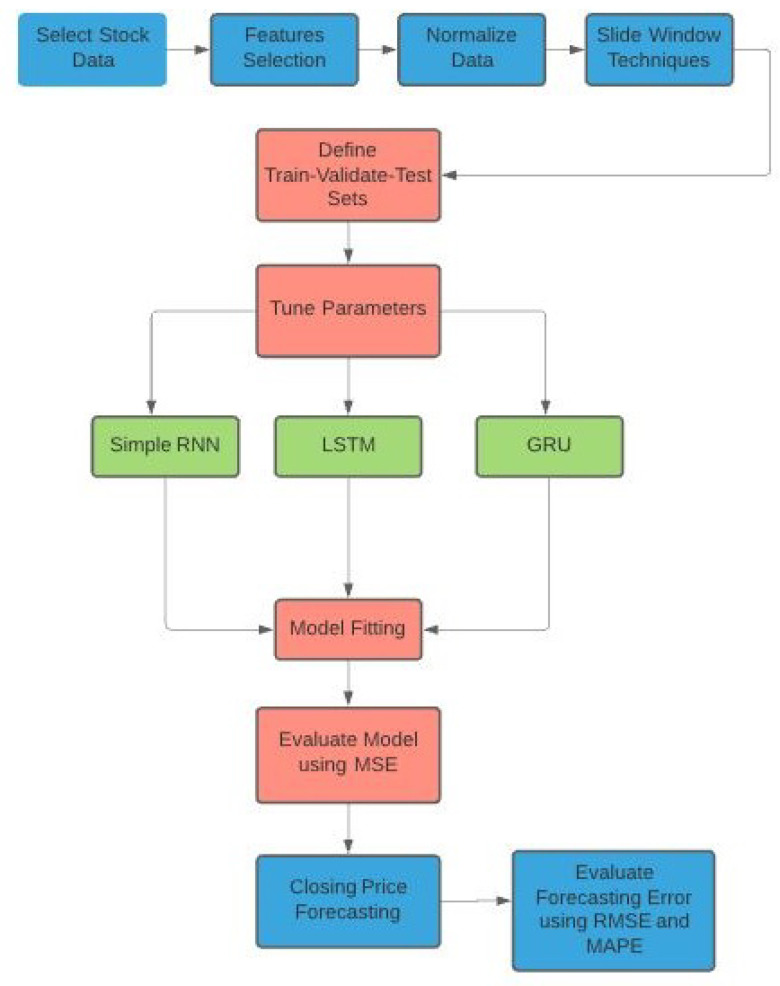
Model design for the RNN forecasting models.

**Table 1 entropy-24-00657-t001:** Summary statistics for the closing price for the eight stock market indices.

Index	Count	Mean	Minimum	Maximum	Standard Deviation
IXIC NYA N100 GDAXI NGE J580.J0 N225 000001.SS	3375 3375 3424 3375 2054 2324 3275 3255	5005.10 10,046.22 838.54 10,046.22 28.15 36,946.57 15,942.95 2859.56	1268.64 4226.31 434.61 4226.31 7.88 20,716.48 7054.98 1706.70	14,138.78 16,590.43 1248.14 16,590.43 65.48 48,467.67 30,467.75 5497.90	2897.71 2457.78 179.82 2457.78 17.79 6700.48 5638.76 574.05

**Table 2 entropy-24-00657-t002:** Summary statistics for the closing price of the six currency exchange rates.

Currency	Count	Mean	Minimum	Maximum	Standard Deviation
ZAR/USD NGN/USD GBP/USD EUR/USD RMB/USD JPY/USD	3488 3487 3487 3474 584 3474	0.097 0.0063 1.49 1.25 0.14 0.01	0.05 0.002 1.14 1.03 0.13 0.007	0.15 1.00 2.03 1.59 0.15 0.01	0.028 0.02 0.18 0.13 0.004 0.001

**Table 3 entropy-24-00657-t003:** Full specification of the parameters of the simple RNN, LSTM and GRU models for the stock market indexes and currency exchange rates.

Parameters	Simple RNN	LSTM	GRU
Learning Rate Number of neurons Number of layer Batch_size Activation function Number of epoch Recurrent_dropout Optimizer Weight Decay Dropout	0.0016 16 3 64 tanh 100 0.3 Adam None None	0.02 32 3 300 tanh 150 None Adam None None	0.0114 128 3 300 tanh 150 None Adam None 0.1801

**Table 4 entropy-24-00657-t004:** Root mean square error for the univariate RNN, LSTM and GRU models, in sample (train and validation sets) and out-of-sample forecasting (train set) of the eight stock market indexes.

RMSE	RNN			LSTM			GRU		
	Train	Valid	Test	Train	Valid	Test	Train	Valid	Test
IXIC	0.0032	0.0049	0.0217	0.0037	0.0050	0.0363	0.0044	0.0042	0.0131
NYA	0.0089	0.0072	0.0167	0.0096	0.0071	0.0155	0.0122	0.0178	0.0241
N100	0.01297	0.0098	0.0164	0.0137	0.0104	0.0161	0.0141	0.0141	0.0172
GDAXI	0.0094	0.0103	0.0148	0.0106	0.0097	0.0158	0.0099	0.0126	0.0152
NGE	0.0128	0.0045	0.0057	0.0144	0.0054	0.0067	0.0244	0.0071	0.0049
J580.JO	0.0154	0.0181	0.0218	0.0187	0.0224	0.0311	0.0232	0.0140	0.0227
N225	0.0092	0.0096	0.0140	0.0098	0.0111	0.0151	0.0105	0.0100	0.0138
000001.SS	0.0071	0.0084	0.0101	0.0148	0.0083	0.0101	0.0143	0.0245	0.0133

**Table 5 entropy-24-00657-t005:** Root mean square error for the multivariate RNN, LSTM and GRU models, in sample (train and validation sets) and out-of-sample forecasting (train set) of the eight stock market indexes.

RMSE	RNN			LSTM			GRU		
	Train	Valid	Test	Train	Valid	Test	Train	Valid	Test
IXIC	0.0027	0.0047	0.0407	0.0034	0.0154	0.1233	0.0036	0.0049	0.0222
NYA	0.0048	0.0041	0.0094	0.0048	0.0068	0.0196	0.0084	0.0038	0.01050
N100	0.0094	0.0079	0.0122	0.0080	0.0106	0.0179	0.0113	0.0082	0.0111
GDAXI	0.0064	0.0111	0.0130	0.0078	0.0079	0.0149	0.0078	0.0073	0.0106
NGE	0.0103	0.0044	0.0071	0.0099	0.0082	0.0074	0.0246	0.0052	0.0047
J580.JO	0.0101	0.0191	0.0165	0.0066	0.0082	0.0142	0.0198	0.0161	0.0128
N225	0.0059	0.0064	0.0126	0.0053	0.0067	0.0170	0.0069	0.0067	0.0089
000001.SS	0.0093	0.0064	0.0072	0.0084	0.0036	0.0058	0.0103	0.0106	0.0057

**Table 6 entropy-24-00657-t006:** Root mean square error for the univariate RNN, LSTM and GRU models, in sample (train and validation sets) and out-of-sample forecasting (train set) of the six currency exchange rates.

RMSE	RNN			LSTM			GRU		
	Train	Valid	Test	Train	Valid	Test	Train	Valid	Test
ZAR/USD	0.0139	0.0079	0.0064	0.0158	0.0102	0.0079	0.0224	0.0183	0.0199
NGN/USD	0.0322	0.0249	0.0092	0.0319	0.0319	0.0075	0.0253	0.0548	0.0007
GBP/USD	0.0123	0.0133	0.0119	0.0145	0.0215	0.0204	0.0191	0.0101	0.0123
EUR/USD	0.0236	0.0104	0.0091	0.0228	0.0112	0.0102	0.0266	0.0189	0.0127
RMB/USD	0.0219	0.0237	0.0363	0.0216	0.0266	0.0333	0.0225	0.0199	0.0199
JPY/USD	0.0194	0.0096	0.0074	0.0202	0.0118	0.0097	0.0245	0.014	0.0087

**Table 7 entropy-24-00657-t007:** Root mean square error for the multivariate RNN, LSTM and GRU models, in sample (train and validation sets) and out-of-sample forecasting (train set) of the six currency exchange rates.

RMSE	RNN			LSTM			GRU		
	Train	Valid	Test	Train	Valid	Test	Train	Valid	Test
ZAR/USD	0.0096	0.0074	0.0083	0.0145	0.0087	0.0070	0.0216	0.015	0.0336
NGN/USD	0.0247	0.0152	0.0159	0.0201	0.0422	0.0536	0.0266	0.0273	0.1534
GBP/USD	0.0094	0.0134	0.0136	0.0089	0.0211	0.0232	0.0165	0.0071	0.0107
EUR/USD	0.0213	0.0118	0.0116	0.0202	0.0097	0.0083	.0250	0.0189	0.0141
RMB/USD	0.0248	0.0188	0.0278	0.0195	0.0269	0.0479	0.0227	0.0178	0.0283
JPY/USD	0.0166	0.0096	0.0069	0.0163	0.0118	0.0102	0.0247	0.0233	0.01292

## Data Availability

The data is available from the corresponding author of this paper.

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
