# Peer review of "Neural Networks for Financial Time Series Forecasting"

_entropy, 2022, doi:10.3390/e24050657_

Round 1

Reviewer 1 Report

Dear Authors
Initially, I thank you for the opportunity to read a well-constructed work and application to society. I realize that this article has already been submitted before. Observing the guidelines issued by the reviewers previously, I can see that the authors responded to the suggestions and implemented them in the body of the work. Thus, not observing other opportunities for improvement, I believe that the article is in minimal conditions for publication. Therefore, my opinion is favorable for the acceptance.
Regards,

Author Response

We thank the reviewer for taking the time to read and evaluate our manuscript.

Author Response

This manuscript is a resubmission of an earlier submission. The following is a list of the peer review reports and author responses from that submission.

Round 1

Reviewer 1 Report

The article aims to predict the closing price of data from eight stock market indices using the RNNs models and their variants.

The EMH's weak form (See Fama (1970) and Fama (1991)) holds that the market is efficient at reflecting all available public information. Market returns are independent and therefore do not help predict future returns. This means that no investor can obtain abnormal returns only through the analysis of historical series. This implies the unpredictability of the Market.

There are, empirical evidence signals the inefficiency in the behavior of stock prices and stock indexes, mainly on issues such as information asymmetry, anomalies (calendar effect, momentum effect, firm size effect, etc), long memory  and the occurrence of black swans by the market dynamics itself.

Still, i particularly am not convinced about the predictability of the NYSE, NASDAQ, Euronext 100, GDAXI,SSE Composite index, Nikkei 225 (N225) , Global X and FTSE/JSE markets treated on paper.

1 – Considering that the covariates used in the multivariate models are correlated with each other. How was collinearity treated by the authors ?

2  – If we change the data partition to a rule 50% -50% or 40% -60% of the results found by the authors remain?

3 – Is it possible to display the evolution of actual values ​​and predicted values ​​for all predicted indices?

Reviewer 2 Report

Dear Authors

Initially, I congratulate the authors for the research work presented. The results contribute to the discussion of the topic. However, we have some observations and suggestions for the authors, in order to present an opportunity for improvement:

As the authors state in the introduction, the stock market is one of the most dynamic and valuable sectors that play a vital role.
in the economic development of a country. This theme has been the object of study in several areas. I believe the authors can take a small approach, such as in the field of operations research, as multicriteria methods applied in the stock market. Thus, it could reinforce the argument that RNN is superior. In this sense, I suggest the authors consider evaluating the following texts:

Basilio, M.P., de Freitas, J.G., Kämpffe, M.G.F. and Bordeaux Rego, R. (2018), "Investment portfolio formation via multicriteria decision aid: a Brazilian stock market study", Journal of Modelling in Management, Vol. 13 No. 2, pp. 394-417. https://doi.org/10.1108/JM2-02-2017-0021

Radojicic, Dragana, Nina Radojicic, and Simeon Kredatus. "A Multicriteria Optimization Approach for the Stock Market Feature Selection." Computer Science and Information Systems 18.3 (2021): 749-69.

Peng, Hong‐gang, Zhi Xiao, Jian‐qiang Wang, and Jian Li. "Stock Selection Multicriteria Decision‐making Method Based on Elimination and Choice Translating Reality I with Z‐numbers." International Journal of Intelligent Systems 36.11 (2021): 6440-470

When reading the introduction, the problem issue and objectives were not clear.
The authors did not indicate the link to access the database used. This is important so that other researchers can use the same data and can reproduce the experiment.
Another point the script indication used to run the tests.

Regarding the conclusion, the authors can explore a little more the contribution of the presented method and its advantages or disadvantages in relation to others.

Regarding written language, I suggest a small review.

Reviewer 3 Report

The research piece assesses the stock market closing price predictive performance (RMSE and MAPE criteria) of three artificial neural network models (RNNs, LSTMs and GRUs) on 8 stock market indices. The authors forecast future closing prices exploiting time series data at a daily frequency (between 6 and 11 years?) for high, low, open, volume and closing price (5 features or inputs), to forecast the future behaviour of the stock market index closing price over a five-period (?) window. They find GRUs  (and RNNs) to perform better than LSTMs.

Overall, the piece is nicely and concisely written, with a general introduction to neural network models, and an application to time series data forecasting. Yet there are important shortcomings:

  • Typically stock market index closing prices are non-stationary, making them unsuitable for time series forecasting exercises. Have the authors checked that over the time window considered for the 8 indices, closing prices are indeed stationary? If they are not, they should report the relevant statistics and use returns instead, which have been found in the literature more likely to be stationary.
  • The authors start correctly motivating the contribution from the perspective of the many possible factors (features) that may influence the evolution of closing stock market index prices, and yet only exploit 5 features in their models. It is unclear whether in such low-dimensional state space (and in a stationary environment), univariate time series forecasting models (e.g. based on A/BIC criteria to select the optimal number of lags) do not perform better. Hence, when assessing performance of statistical learning methods, univariate time series models should be in the list of model alternatives.
  • Although RMSE and MAPE levels are reported, there is no statistical test reported (e.g. Diebold and Mariano test statistic) that allows to compare them formally: are the observed differences (reported in Table 3) statistically significant?
  • Robustness: Are the results reported in Table 3 robust to changes in the forecasting horizon (e.g. from 5 to 1 or 10 periods ahead)? Are they robust to imposing equally sized architectures (in terms of total number of units)?
  • English proofreading necessary, e.g. line 404 ‘The aim of this paper was to [access] the performance of the neural networks models, 405 particularly the recurrent neural network models, for stock market forecasting.’ In the conclusion should read instead [assess]?

Reviewer 4 Report

This study uses some neural network methods to forecast stock prices. The methods used are not new, even to financial markets. Usually, we forecast stock returns instead of prices. Also, the comparisons are between three NN methods, rather than between NN methods and traditional forecasting methods. Therefre, the conclusions are not convincing.
